

# A randomized control trial feasibility evaluation of an *m*Health intervention for wheelchair skill training among middle-aged and older adults

Edward M. Giesbrecht[1] and William C. Miller[2]

[1] Department of Occupational Therapy, University of Manitoba, Winnipeg, MB, Canada
[2] Department of Occupational Therapy and Occupational Science, University of British Columbia, Vancouver, BC, Canada

## ABSTRACT

**Background:** Providing mobility skills training to manual wheelchair (MWC) users can have a positive impact on community participation, confidence and quality of life. Often such training is restricted or not provided at all because of the expense of, and limited access to, occupational and physical therapists before and after discharge. This is particularly true among middle-aged and older adults, who often have limited access to rehabilitation services and require more time to learn motor skills. A monitored MWC skills training home program, delivered using a computer tablet (*m*Health), was developed as an alternative approach to service delivery. The purpose of this study was to evaluate the feasibility of implementing this *m*Health MWC skills training program among middle-aged and older adults.

**Methods:** A 2 × 2 factorial design randomized controlled trial (RCT) was used to compare the *m*Health intervention and control groups, with additional wheeling time as a second factor. Community-dwelling MWC users aged 55 and older, who had used their MWC for less than two years and propelled with two hands, were recruited. Feasibility outcomes related to process, resources, management and treatment criteria were collected.

**Results:** Eighteen participants were recruited, with a retention rate of 94%. Mean (±SD) duration for the first and second in-person training sessions were 90.1 ± 20.5 and 62.1 ± 5.5 min, respectively. In the treatment group, 78% achieved the minimum amount of home training (i.e., 300 min) over four weeks and 56% achieved the preferred training threshold (i.e., 600 min). Trainers reported only seven minor protocol deviations. No tablets were lost or damaged and there was one incident of tablet malfunction. No injuries or adverse incidents were reported during data collection or training activities. Participants indicated 98% agreement on the post-treatment benefit questionnaire.

**Discussion:** Overall, the study protocol enabled implementation of the intervention in a safe, efficient and acceptable manner. Participant recruitment proved to be challenging, particularly gaining access to individuals who might benefit. Resource issue demands were acceptable for administration of the intervention; data collection was more time-consuming than anticipated but could be reduced with minor revisions. Participant retention and home program treatment adherence was high; both participant and trainer burden was acceptable. Treatment group

Corresponding author
Edward M. Giesbrecht,
ed.giesbrecht@umanitoba.ca

participants reported a positive experience and clinical benefits from training program. The findings suggest a full-scale RCT evaluating the clinical impact of the Enhancing Participation In the Community by improving Wheelchair Skills (EPIC Wheels) intervention is warranted, provided the recruitment issues are addressed through collaborative partnerships and active recruitment strategies.

## INTRODUCTION

For those with a mobility limitation, an appropriately prescribed manual wheelchair (MWC) can improve participation (*Chaves et al., 2004*), community mobility, and quality of life (*Requejo, Furumasu & Mulroy, 2015*; *Winkler et al., 2008*) while reducing caregiver burden and personal assistance costs (*Cohen & Perling, 2015*). However, the benefits of MWC use are dependent upon the user's ability and confidence to operate the device safely and effectively, in order to navigate the environmental barriers and obstacles they encounter (*Mortenson et al., 2012*; *Phang et al., 2012*). These wheelchair mobility skills range from basic and essential ones, such as propulsion and transfers, to more complex, such as managing doors, ramps and curbs. There is evidence that providing specific skills training to MWC users has a positive impact on confidence (*Sakakibara et al., 2013*), participation (*Kilkens et al., 2005*; *Mortenson et al., 2011*), mobility (*Hoenig et al., 2005*; *Lemay et al., 2012*) and quality of life (*Hosseini et al., 2012*). Unfortunately, many MWC users receive limited training, often focusing on only the most basic skills (*Kirby et al., 2015*; *Nelson et al., 2010*). This is particularly true among middle-aged and older adults (*Karmarkar et al., 2009*) who often have limited access to rehabilitation services (*Sanford et al., 2006*; *Tousignant et al., 2007*). Lack of time and access to rehabilitation services are typically implicated for this lack of training (*Best et al., 2016*; *Routhier et al., 2012*). While in hospital, occupational and physical therapists often focus on interventions that target the transition home, and older adults may not receive their permanent MWC until after discharge (*Kirby et al., 2015*). Once in the community, access to out-patient services is often restricted and older adults report challenges in attending out-patient therapy due to transportation access, cost, and scheduling issues (*Giesbrecht et al., 2014*; *Sanford et al., 2006*; *Tousignant et al., 2006*).

Some attempt has been made to deliver MWC skills training in a comprehensive and structured manner, most notably the Wheelchair Skills Training Program (*MacPhee et al., 2004*; *Sakakibara et al., 2013*). In most cases, a clinician with wheelchair expertise delivers the training program to an individual in 5–10 sessions of 30–60 min duration, over a period of two to four weeks (*Kirby et al., 2015*; *Routhier et al., 2012*). The number of training sessions required typically increases with older adults, where age-related changes impact motor learning processes (*Bonaparte, Kirby & MacLeod, 2004*; *Voelcker-Rehage, 2008*). This approach to training requires the clinician to be available for multiple training sessions; the trainee to coordinate appointments with the trainer;

and the MWC user to arrange and pay for transportation to these appointments. Furthermore, training in a clinic setting requires the MWC user to translate those mobility skills learned into the context of their own environment, leading to apprehension around attempting to navigate obstacles they encounter in their community (*Walker et al., 2010*). Alternately, training can also occur in a community setting (*Best et al., 2005*). This approach allows practice in the context of use, but still requires coordinated appointments and the trainer travelling to the MWC user's home and community locations for multiple sessions, which is cost-prohibitive.

In response to this service gap, alternative and cost-effective approaches need to be developed to improve access to, and effectiveness of, wheelchair skills training for middle-aged and older adults. We developed a telerehabilitation MWC skills training intervention primarily delivered as a home program using a computer tablet. In the field of rehabilitation, use of *mobile* devices such as a tablet or smart phone (i.e., *m*Health) (*World Health Organization, 2011*) is growing (*Sama et al., 2014*). This approach has been useful with interventions targeting health literacy (*O'Connor, Farrow & Hatherly, 2014*; *Watkins & Xie, 2014*), self-management (*Murray, 2012*), and health behavior change (*Webb et al., 2010*); however, this is the first application of *m*Health we are aware of that addresses training of a motor skill, specifically MWC mobility. Enhancing Participation In the Community by improving Wheelchair Skills (EPIC Wheels) combines two in-person training sessions and a four-week home program delivered via a tablet and monitored by the trainer. EPIC Wheels was developed collaboratively with middle-aged and older adult MWC users, caregivers and clinicians involved in skills training (*Giesbrecht et al., 2014*), and explicitly incorporates principles of adult learning (*Knowles, 1980*) and self-efficacy theory (*Bandura, 1997*). Social cognitive theory proposes that self-efficacy is influenced most by successful performance (mastery experience), as well as observing success in comparable others (vicarious experience), encouragement from valued others (verbal persuasion), and appropriate interpretation of affective and physical experiences (*Bandura, 1997*). The tablet program includes 158 video-based components that provide instruction and demonstration, as well as training activities and games. User activity is uploaded to a secure server, where the trainer can monitor progress; the MWC user and trainer can exchange voice messages using the EPIC Wheels application. This intervention addresses some of the salient barriers to traditional MWC skills training. It incorporates evidence-based content as well as instruction and monitoring by a wheelchair expert, but limits the burden of time for the trainer and travel for the MWC user. In addition, MWC users can practice in their actual context of use (*Walker et al., 2010*) and have control over the pace, sequence and frequency of training (*Jordan et al., 2010*; *Shaughnessy & Resnick, 2009*).

Following the development phase of the EPIC Wheels program (*Giesbrecht et al., 2015*), the next step was to evaluate administration and impact of the intervention with the target population. Implementation and evaluation of telerehabilitation interventions is still in an emergent phase (*Kumar et al., 2013*) and there are a number of factors that could potentially impact feasibility in practice (*Kairy et al., 2009*). Adherence is essential to the success of any home program intervention (*Jordan et al., 2010*;

*Shaughnessy & Resnick, 2009*). A critical evaluation factor is whether participants will sustain engagement in their home training with limited in-person contact and whether this training can be completed safely (*Altilio et al., 2015*). Acceptability and uptake of *m*Health by middle-aged and older adults is a potential barrier to EPIC Wheels implementation; low self-efficacy and technology-related anxiety may contribute to this reluctance (*Chen & Chan, 2011*; *Laguna & Babcock, 2000*). Recent data suggests that 27% of Americans over 65 years own a tablet or e-book reader and 18% own a smartphone; however, this population is still more reluctant than their younger counterparts to adopt this type of technology and only 47% have high-speed Internet access in their home (*Smith, 2014*). Assistive technology adoption theory suggests that reliable and consistent device performance, as well as technical support when required, is critical to user uptake particularly among older adults (*Venkatesh, Thong & Xu, 2012*).

The impact of a rehabilitation intervention is often evaluated in a clinical trial. A randomized controlled trial (RCT) provides the strongest level of evidence for treatment efficacy, but can be expensive, time-consuming and difficult to implement. Recruitment and retention are among the most significant challenges to conducting an RCT, and present a threat to study feasibility with our target population (*Hubbard et al., 2015*). Middle-aged and older adults, particularly MWC users, are more likely to decline participation and discontinue treatment due to co-morbidities and complex health conditions (*McMurdo et al., 2011*; *Nary, Froehlich-Grobe & Aaronson, 2011*). The literature reports increased prevalence of depression, apathy, cognitive impairment and low self-efficacy among older adults, contributing to lower recruitment and retention rates (*Corcoran et al., 2016*). Potential participants may be reluctant to participate or attend all appointments if they must travel frequently, especially if the location is inconvenient (*Page & Persch, 2013*) and there is insufficient reimbursement for travel costs (*Blanton et al., 2006*). MWC users are particularly vulnerable because of financial restrictions (*Nary, Froehlich-Grobe & Aaronson, 2011*), dependence on para-transit services, and the additional demands placed on caregivers to coordinate travel (*Giesbrecht, Miller & Woodgate, 2015*; *Nary, Froehlich-Grobe & Aaronson, 2011*). In addition, targeting a population defined by a functional limitation (i.e., restricted mobility), rather than a diagnostic condition, makes it difficult to identify and target recruitment venues (*Nary, Froehlich-Grobe & Aaronson, 2011*).

Therefore, before evaluating the EPIC Wheels program in a large-scale RCT, we undertook a feasibility study to confirm that the proposed design would be sufficiently robust and identify whether further changes would be required before moving forwards (*Lancaster, 2015*). The purpose of this study was to evaluate the feasibility of implementing an RCT with an *m*Health MWC skills training program among middle-aged and older adults. Specifically, we considered four broad components of feasibility (*Thabane et al., 2010*). *Process* outcomes evaluated participant recruitment and retention, as well as treatment adherence. *Resource* issues related to the viability of collecting data and administering the intervention, and the resultant burden on testers, trainers and

participants. *Management* issues considered participant processing time, protocol fidelity and equipment reliability. *Treatment* issues focused on participant safety and perceived benefit.

## MATERIALS AND METHODS

### Study design

One value of a RCT is the strength of design to establish that benefit is due to a *specific* treatment rather than treatment generally (e.g., a placebo benefit) through the use of a control group (*Moffett, 1991*). In the case of the EPIC Wheels intervention, several confounding factors could potentially impact mobility-related outcomes. First, an intervention focused on wheelchair mobility could motivate participants to attend to wheelchair activity and use, much like New Year's Day can initiate increased exercise activity for some individuals. Second, interaction with a trainer might increase motivation and elevate mood, potentially increasing participants' attention to wheeling activity. Third, using a tablet device could influence participants' engagement because of the novel delivery method. Consequently, the control group intervention was configured to address these specific variables by closely matching the number, duration and type of contacts with the trainer and providing a parallel intervention via a computer tablet (*Portney & Watkins, 2009*). There is substantial research related to use of cognitive and commercial computer games to effect clinical benefits in rehabilitation, although the results are generally task-specific and the generalizability to functional benefits is more equivocal (*Kueider et al., 2012*; *Pichierri et al., 2011*). Thus, cognitive training using computer games provided a conceivable intervention to improve wheelchair mobility skills, thereby achieving some degree of clinical equipoise. However, because the treatment group participants were required to perform specific tasks *with* their MWC (which the control group participants were not), an argument could be made that any benefit realized might simply be the result of increased MWC use, rather than the specific intervention. To address this, an expectation of increased wheelchair use (i.e., "extra wheeling") was introduced as a second intervention variable, such that participants would be randomly assigned to treatment/control and extra-wheeling/no extra-wheeling. This additional factor would enable between-group comparisons for the primary factor and secondary factor, as well as a potential interaction effect (*Portney & Watkins, 2009*).

A randomized control trial was used employing a $2 \times 2$ factorial design. Participants were allocated into four groups (i.e., EPIC Wheels with and without extra wheeling; cognitive training with and without extra wheeling) using a 1:1:1:1 allocation ratio. A computer-generated randomization process, with undisclosed block size and stratified by site, was used to ensure comparable group size while still masking assignment. After enrolment, baseline data was collected and the participant was assign to a group. An initial in-person training session was scheduled with the group-specific trainer, followed by a four-week home training program. A second in-person training session was conducted at the mid-way point (i.e., after two weeks of home training) and the tester re-administered outcome measures after the home

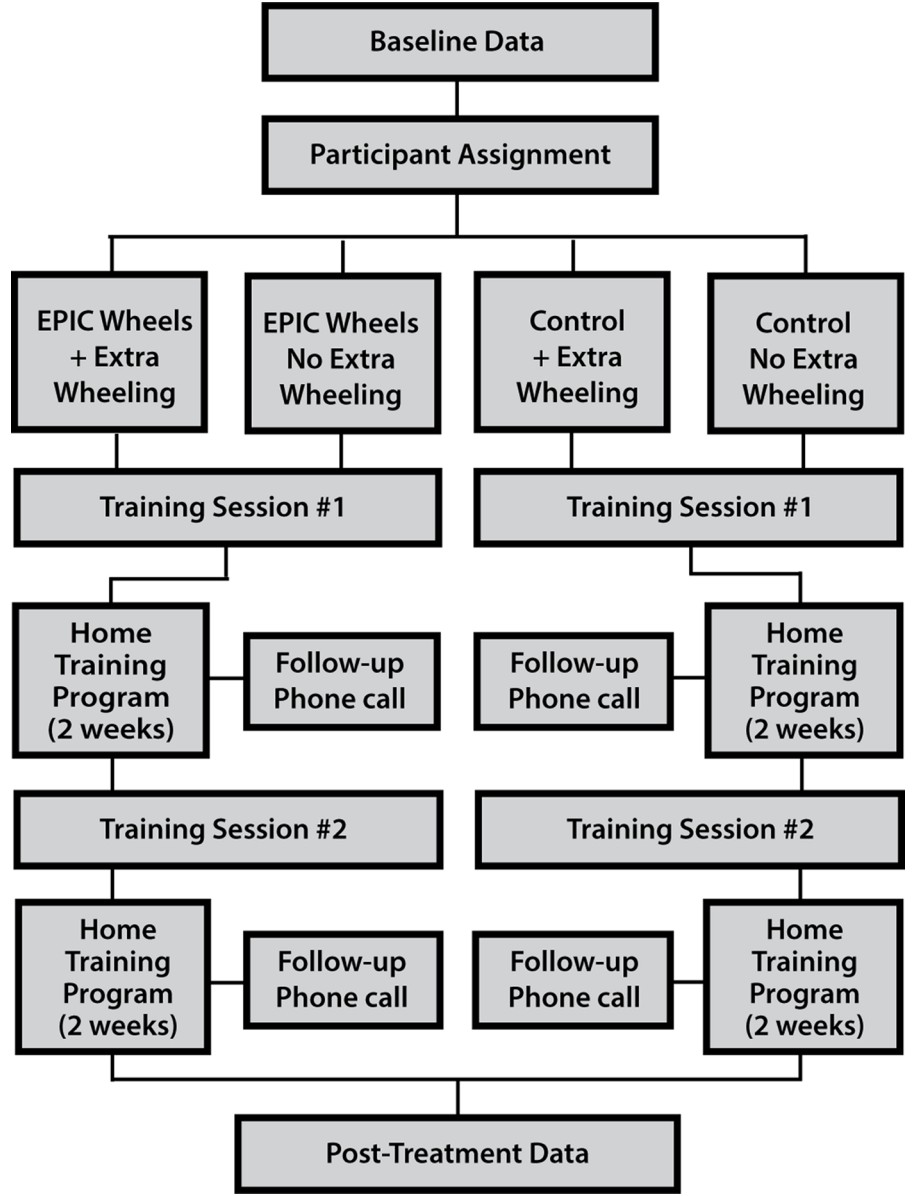

**Figure 1 Randomized control trial study design.**

program was complete (see Fig. 1). Several strategies were used to reduce the risk of bias. Research assistants collecting data were blinded to the group assignment and participants were encouraged not to discuss their training program. In addition, the trainer for the treatment and control groups were separate individuals. Additional details on methodology are available in a previously published protocol paper (*Giesbrecht et al., 2013*).

This study was approved by the Research Ethics Boards at the University of Manitoba (Approval #: H2012:330) and the University of British Columbia (Approval #: H12-02043) and was registered as a clinical trial (ClinicalTrials.gov #NCT01740635). Participants provided written informed consent at the time of enrolment.

## Participants

Community-dwelling MWC users living in two Canadian cities (Winnipeg and Vancouver) were recruited on a volunteer basis between March 2013 and March 2016. Initially, to optimize the impact of the treatment, individuals with less than one year of MWC use were recruited. Novice users are still developing routines and patterns of wheelchair use and potentially more amenable to adapting their mobility techniques (*Coolen et al., 2004*). Over the course of the study, this criterion was adjusted to <2 years of use and then subsequently removed altogether to enhance recruitment efforts. Initially, participants were restricted to those ≥55 years old, but this criterion was also modified to ≥50 years old to address recruitment issues. Additional criteria included living in the community and being able to self-propel a MWC at least 1 h/day inside and outside of the home. Inability to communicate or complete study questionnaires in English, receiving concurrent mobility training, and health conditions that contraindicated training were exclusion criteria. Participants were encouraged, but not required, to have a caregiver present during the in-person and home training sessions.

While the principal focus of the study was feasibility evaluation, a sample size calculation was undertaken to provide an estimate for recruitment needed to detect a statistically significant difference between groups on the primary clinical outcome (i.e., Wheelchair skill test-capacity). Using a data subset for older adult MWC users, a mean change of 9.3% (SD = 9.5%; $\rho$ = 0.49) with α set at 0.10 indicate eight participants per group would provide 90% power using ANCOVA for an RCT design; after conservatively adjusting for a 25% attrition rate, the total number of participants targeted was 44 (*Borm, Fransen & Lemmens, 2007*).

## Procedures

Participants attended a 2 h in-person session with their trainer. This session included an orientation to the tablet and home training program. They took the tablet home with them and were instructed to perform a minimum of 75 min of home training per week, but encouraged to attempt 150 min/week, training 1–2 sessions/day, 15–30 min in length, at least five days per week. Participants returned for a second 1 h session two weeks later and then continued with their home training program for an additional two weeks. To prevent attrition, the trainer contacted participants by telephone at the end of weeks 1 and 3 to address any issues, provide encouragement and promote adherence (*Jette et al., 1998*).

Participants assigned to the extra wheeling groups were instructed to spend 75 min per week in unstructured wheelchair wheeling (i.e., not typical of their daily activities); this was *in addition to* time spend with in tablet-based training.

### Treatment group

All treatment sessions were conducted according to a written protocol. EPIC Wheels (treatment group) trainers provided 1 h of instruction, which included demonstration, practice and feedback. Skills were progressed according to the participant's level of

function. At the end of 1 h, trainers recommended skills to focus on during home training program based on ability, safety, and relevance.

The EPIC Wheels home program included a comprehensive, structured library of educational material and training activities organized in a hierarchy from simple to complex, delivered using a 10″ Android computer tablet. Training was provided in a multi-media format with illustrations and videos, allowing detailed step-by-step guidance and demonstration. Age-appropriate actors of both sexes were used to demonstrate skills in the videos, providing vicarious reinforcement consistent with principles of social cognitive theory (*Bandura, 1997*). Participants also received a small mobile Internet device that was pre-configured to provide Wi-Fi connectivity for voicemail and data transfer/ update capability; however, the EPIC Wheels program could operate independently without Internet access for training purposes. A small platform supported the tablet on the participant's lap for in-wheelchair use, using a simple strap around the subject's thighs for stability. A "Progress" icon provided daily updates on the number of minutes practiced per week, as well as percentage of the weekly goal, to reinforce adherence. This progress information was a subset of the data uploaded regularly to the trainer's website. Messages could be exchanged between subject and trainer using a voicemail function in the EPIC Wheels program on their tablet and website, respectively. Trainers monitored each participant's training activity by regularly visiting the website. The trainer could use this information to identify adherence issues and address these through voicemail, the bi-weekly follow-up phone call, or at the second training session.

### Control group

Comparably, the control group participants met with their trainer for two sessions, each following a written protocol, and received bi-weekly follow-up phone calls. They received a tablet pre-loaded with nine commercially available games related to problem solving; word, math and memory activities; and fine motor skills. Each training session was administered using a separate protocol and checklist. Session one focused on benefits of engaging in computer game training and the potential impact wheelchair use might have on mobility. Session two focused on participants' current activities in the community and providing information on how to address barriers encountered.

### Data collection

At baseline, descriptive participant characteristics were documented. A total of eight clinical outcome measures were collected at baseline and post-treatment (i.e., immediately following the four-week home training) following a structured written protocol, including recording of administration time. The clinical outcomes, and their associated constructs, are listed in Table 1; additional detail is available in a published protocol paper (*Giesbrecht et al., 2013*).

## Feasibility indicators

A priori measurement criteria were established for the feasibility indicators (see Table 2); these criteria served as hypotheses to establish study feasibility. Indicators evaluated as

**Table 1 Clinical outcome measures.**

| Clinical construct | Outcome measure |
|---|---|
| Skill capacity* | Wheelchair skills test-capacity (WST-C) |
| Safety | Wheelchair skills test-safety (WST-S) |
| | Wheeling while talking (WWT) test |
| Self-efficacy | Wheelchair use confidence scale (WheelCon) |
| Participation in occupation | Wheelchair outcome measure (WhOM): Indoor and outdoor subscales |
| Mobility | Life-space assessment (LSA) |
| Health related quality of life | Health utilities index (HUI) |

**Note:**
* Primary clinical outcome measure.

"achieved" were considered sufficiently robust requiring little or no adaptation. Indicators evaluated as "revise" would need to be addressed before proceeding.

*Process* components reflected the feasibility of the various steps involved in undertaking the study (*Thabane et al., 2010*). Site coordinators made on-going documentation of participant inquiries, responses, recruitment, appointment scheduling and attendance. The threshold for retention was set at 80%, as a 20% loss of data can present a threat to study validity (*Brueton et al., 2013*). Adherence was assessed via uploaded tablet usage data (i.e., time spent in home training). This included training session frequency, minutes accessing program components (e.g., instructional videos, training activities, and games) and time spent training without the tablet (manually inputted on the tablet). Weekly and total study period totals were compared against the minimum (75 and 300 min, respectively) and preferred (150 and 600 min, respectively) practice guidelines.

*Resource* components related to time and budget demands (*Thabane et al., 2010*). Testers completed a data collection protocol checklist at baseline and post-treatment, and documented administration time. In particular, administer time and sensitivity to change for the health utilities index (HUI) was of interest because of its potential application for cost-benefit analysis in future studies. Trainers completed protocol checklists at each of the two training session and documented administration time.

*Management* components dealt with personnel, equipment and data issues (*Thabane et al., 2010*). Site coordinators documented the time between data collection and training appointments, and any equipment issues during training. Trainers indicated any protocol deviations on the checklists and completed a trainer post-treatment evaluation form after EPIC Wheels participants had completed all training.

*Treatment* components related to assessment of safety, dose-specific response, and evaluation of perceived benefit (*Thabane et al., 2010*). Adverse events were documented using the protocol checklists for data collection and in-person training, and via daily tablet prompts during home training. The mean EPIC Wheels treatment group change score was compared to a Minimum Detectable Change of 3.0% for the primary outcome (wheelchair skill capacity). The tablet usage data was used to explore potential benefits of higher treatment dosage using the minimum and preferred training time thresholds. Perceived benefit for the EPIC Wheels participants was assessed using a participant

**Table 2 Feasibility indicators, proposed criteria and outcomes.**

| Feasibility component | Indicator | Criteria | Outcome |
|---|---|---|---|
| *Process* | | | |
| Recruitment rate | # of subjects recruited | three subjects/month/site: total of 44 over 8 months | Revise |
| Consent rate | % of subjects consenting | <10% subject refusal | Revise |
| Retention rate | % of subjects with DC2 | Complete data collection for >80% | Achieved |
| Treatment adherence | Attend both training sessions | >85% of subjects | Achieved |
| (EPIC group) | Meet minimum practice time guidelines | >85% of subjects | Achieved |
| (Control group) | Both training sessions conducted | >85% of subjects | Achieved |
| *Resources* | | | |
| Data collection: subject and tester burden | DC1 duration | >85% of subjects complete in ≤2 h | Revise |
| | DC2 duration | >85% of subjects complete in ≤1.5 h | Revise |
| Collection of HUI data | Administration | Mean HUI administration is <10 minutes | Achieved |
| | HUI pre/post score | Statistically significant change pre-post | Achieved |
| Trainer burden | Time spent with subject in training intervention | Mean time ≤2 h for Session 1 and ≤1 h for Session 2 | Achieved |
| *Management* | | | |
| Participant processing time | Time from data collection to treatment | Mean time is ≤10 days at each site | Revise |
| Tablet reliability | Downtime due to technical or mechanical issues | >90% of subjects are not without a tablet for >2 days | Achieved |
| Equipment loss/damage | Tablet is lost/unusable | <2 tablets lost over study | Achieved |
| Treatment administration issues | Post-treatment evaluation form (study trainer) | Any issues identified modifiable without substantial changes to the protocol | Achieved |
| *Treatment* | | | |
| Safety (data collection and training) | Adverse events during assessment or training | No major injuries or adverse events reported | Achieved |
| Safety (home program) | Adverse events during home training | No major injuries or adverse events reported | Achieved |
| Dose level response | Training expectations effect a change score | Minimum practice time guidelines sufficient for a treatment effect | Achieved |
| Perceived benefit | Post-treatment participant questionnaire | >85% of responses will be "strongly agree/agree" | Achieved |

**Notes:**
DC1, baseline data collection; DC2, post-treatment data collection; HUI, health utilities index.

post-treatment questionnaire at the last data collection appointment. To avoid a respondent bias, participants received the questionnaire from the site coordinator (who was not involved in data collection or delivery of the intervention) and completed it independently in a private space.

## Data analysis

For *Process* components, the rates for recruitment, consent, retention and adherence were calculated as frequency counts and percentages. For *Resource* components, frequency counts were used to determine subject and tester burden, and trainer burden was obtained using mean training session duration. For the HUI measure, mean administration time was calculated and HUI scores for EPIC Wheels participants were compared pre- and post-intervention using a paired *t*-test. For *Management* components, simple counts

were used to tabulate days between data collection and treatment initiation, tablet technical issues, and tablet downtime days. The number and nature of protocol administration issues were counted and qualitatively assessed with respect to changes indicated. For *Treatment* components, adverse events (in-person and home training) were tabulated as simple counts. A mean change score on the primary outcome (wheelchair skill capacity) was calculated for EPIC Wheels group completers. In addition, differences in change score between EPIC Wheels participants achieving the preferred (600 min) training dose and those meeting the minimum (300 min) training dose were compared using an independent samples $t$-test. Given that this was a feasibility study, all $t$-tests were conducted with $\propto$ set at 0.10 to ensure a potentially beneficial treatment effect did not go undetected (Type II error). Post-treatment questionnaires responses were treated as simple counts summed to obtain percentages.

## RESULTS

### Process indicators

A total of 18 participants were recruited at two sites (Site 1 $n = 7$; Site 2 $n = 11$) over a period of 36 months, for an average of approximately one participant every two months. A CONSORT flow diagram for recruitment is provided in Fig. 2. A total of 55 individuals were contacted, 18 of who were deemed ineligible. Among the 37 eligible individuals, 12 (32%) were lost to follow-up or died, 7 (19%) declined to participate, and 18 were enrolled for a consent rate of 49%. One participant from the EPIC Wheels group withdrew shortly after enrolment due to an emergent and unrelated health issue. Of the 18 individuals who consented to participate, 17 completed both data collection sessions, for a retention rate of 94%. The mean (±SD) age of participants was 66.1 (±9.5) years, ranging from 50 to 84 years, and were predominantly males ($n = 13$).

Seven of the eight control group participants (88%) and nine of the 10 EPIC Wheels participants (90%) attended both training sessions. For EPIC Wheels participants, a summary of the home training guidelines and performance data is summarized in Table 3 and detailed training activity in Table 4. Among the EPIC Wheels participants who completed the study ($n = 9$), 78% achieved the minimum amount of home training (i.e., 300 min) over the full four weeks and 56% achieved the preferred training threshold (i.e., 600 min). There was considerable variation in total home training time (range 105–1,443 min) as well as type of training EPIC Wheels participants engaged. Specifically, total time engaging in tablet-based training range from 105 to 1,382 min, and time reported as skill training activity without the tablet ranged from 0 to 980 min.

### Resource indicators

Data collection duration data was only available for 13 participants. At baseline, the mean duration was 150.8 ± 48.1 min; three participants (31%) completed testing in ≤2 h. At post-treatment, six participants (46%) finished in ≤1.5 h; the overall mean was 105.7 ± 47.3 min. The HUI administration time was available for ten study participants, with a mean of 9.1 ± 4.2 min. A paired $t$-test comparing pre- and post-intervention HUI scores for EPIC Wheels participants indicated a statistically significant improvement

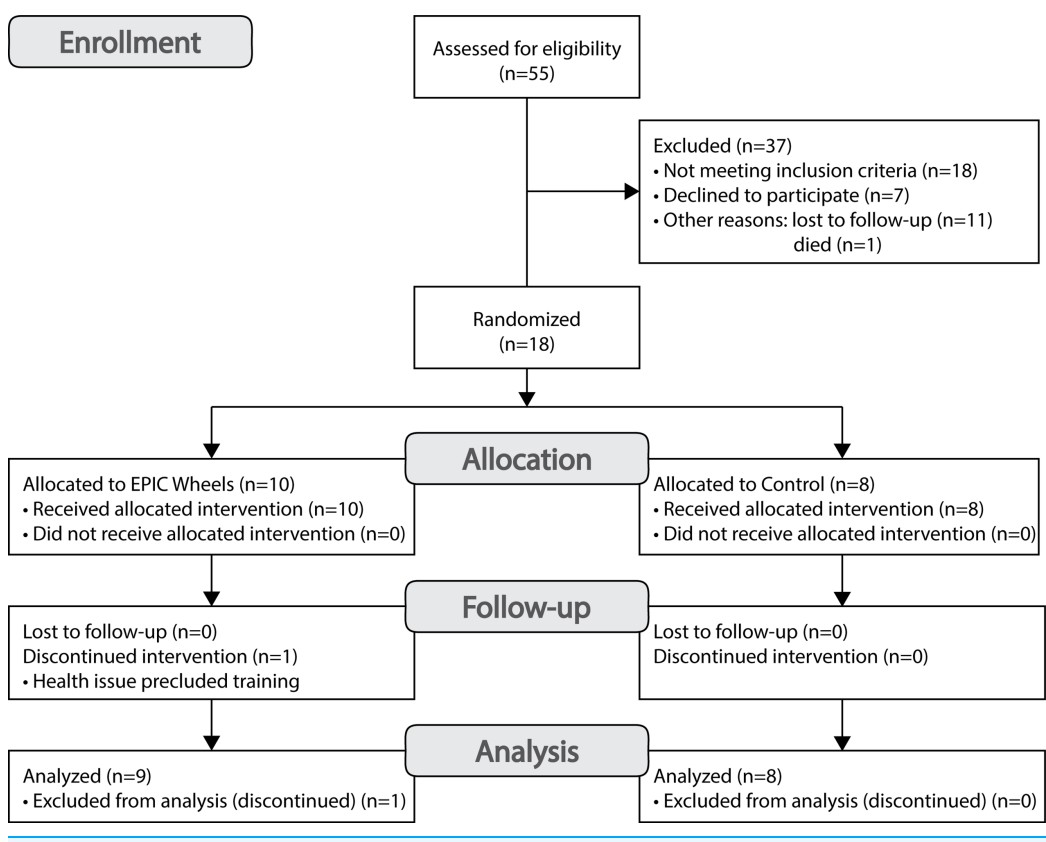

**Figure 2  CONSORT flow diagram for participant recruitment.**

($t = 2.45$, df $= 8$, $p = 0.04$). Data for the first in-person training session was available for 16 participants; all sessions were conducted in less than 2 h ($m = 90.1 \pm 20.5$ min). For the second in-person session ($n = 12$), mean training time was $62.1 \pm 5.5$ min overall with all participants finishing in less than 90 min.

## Management indicators

The mean processing time between study enrolment and initiation of the intervention was $11.3 \pm 6.8$ (range 4–28) days (Site 1: $13.9 \pm 9.0$ days; Site 2: $9.6 \pm 4.8$ days). There was one incident of tablet malfunction (6% of participants) that required replacement for an EPIC Wheels participant, resulting in the participant not having tablet access for several days; this participant was still able to achieve the minimum training threshold. Several participants required some assistance from the study coordinator to resolve voicemail or Wi-Fi connectivity issues; however, none of these situations resulted in participants losing tablet access. No tablets were lost or damaged over the course of the study. Trainer post-treatment protocol evaluation forms were available for 16 participants and indicated no major protocol deviations. A total of seven minor protocol deviations were reported: one participant required a minor wheelchair adjustment prior to initiating training; the timing for follow-up phone calls was modified for one participant; three participants required an additional in-person visit, two for assistance with tablet operation and one for additional care provider spotter training; and two participants had an abbreviated

Table 3 EPIC Wheels home training guidelines and participant results ($n = 7^*$).

| Parameter | Instruction to participant | Minimum | Mean ± SD (range) |
|---|---|---|---|
| Frequency | | | |
| Total days of training | 20–28 days | 20 days | 15.0 ± 5.9 days (8.0–26.0) |
| Days/week training | 5–7 days/week | 5 days | 3.8 ± 1.5 days (2.0–6.5) |
| Intensity† | | | |
| Training duration/day | 15–30 min | 15 min | 45.1 ± 19.9 (10.9–72.2) |
| Dosage | | | |
| Minutes of training/week | 75–150 | 75 min | 183.6 ± 171.0 (26.2–360.8) |
| Total minutes training | 300–600 | 300 min | 734.0 ± 459.5 (104.6–1443.0) |

Notes:
 * Home training data was only available for seven participants.
 † Intensity of treatment was defined as the duration of training time in a single day.

Table 4 EPIC Wheels home program training data for EPIC Wheels participant.

| Participant | Week 1 | | Week 2 | | Week 3 | | Week 4 | | Total | | |
|---|---|---|---|---|---|---|---|---|---|---|---|
| | Tablet[a] | All[b] | Tablet | All | Tablet | All | Tablet | All | Tablet | Other[c] | Grand[d] |
| Site 1 P4 | 46.7 | 182.0 | 54.0 | 319.0 | 64.5 | 300.0 | 57.4 | 402.0 | 222.6 | 980.0 | 1202.6 |
| Site 1 P5 | 218.9 | 279.0 | 318.7 | 374.0 | 232.6 | 263.0 | 148.2 | 248.0 | 918.4 | 245.0 | 1163.4 |
| Site 1 P7 | 228.2 | 228.2 | 399.4 | 399.4 | 84.2 | 84.2 | 0.0 | **0.0** | 711.8 | 0.0 | 711.8 |
| Site 2 P2 | 141.0 | 152.0 | 85.0 | 85.0 | 630.0 | 665.0 | 526.0 | 541.0 | 1382.0 | 61.0 | 1443.0 |
| Site 2 P3 | 214.5 | 265.0 | 60.5 | **60.5** | 50.8 | 90.8 | 86.0 | 171.0 | 411.8 | 175.0 | 586.8 |
| Site 2 P6 | 126.3 | 126.3 | 1.1 | **1.1** | 13.3 | **53.3** | 5.6 | **5.6** | 146.3 | 40.0 | **186.3** |
| Site 2 P7 | 133.9 | 148.9 | 69.9 | **69.9** | 124.5 | 154.5 | 8.0 | 108.0 | 336.3 | 145.0 | 481.3 |
| Site 2 P10 | 495.1 | 550.1 | 61.2 | 121.2 | 0 | **10.0** | 0 | **45.0** | 556.3 | 170.0 | 726.3 |
| Site 2 P11 | 63.9 | **63.9** | 20 | **20** | 1.5 | **1.5** | 19.2 | **19.2** | 104.6 | 0 | **104.6** |
| Mean ± SD | 185.4 ± 133.1 | 221.7 ± 140.9 | 118.9 ± 140.0 | 161.1 ± 157.5 | 133.5 ± 200.0 | 180.3 ± 209.5 | 94.5 ± 169.4 | 171.1 ± 416.0 | 532.2 ± 443.1 | 201.8 ± 304.1 | 734.0 ± 459.5 |

Notes:
 Values in **bold font** did not meet the weekly minimum goal of 75 min or total training period goal of 300 min.
 a Minutes spent on tablet-related training (i.e., watching demo videos, training games and timed training activities).
 b Total minutes of tablet-related training + self-reported non-tablet training time.
 c Total minutes of self-reported non-tablet training time.
 d Grand total of tablet and non-tablet training time.

training session due to a conflicting appointment. Several additions to the protocol were identified: ensuring the Wi-Fi device was activated prior to tablet demonstration; including data logger installation in the task checklist; and providing options for participants who could not tolerate the tablet holder on their lap. All post-treatment evaluations indicated the protocol was clear and 87.5% confirmed the time allocated for administration was reasonable.

## Treatment indicators

There were no injuries or adverse incidents during any data collection sessions, training sessions, or home training. The mean change score for wheelchair skill capacity among all EPIC Wheels treatment group participants was 5.3 ± 7.1% and 4.6 ± 7.1%

**Table 5 EPIC Wheels group post-treatment questionnaire results (_n_ = 9).**

| Item | Strongly disagree | Disagree | Agree | Strongly agree |
|---|---|---|---|---|
| Receiving wheelchair skills training is valuable or important for me | | | | 9 |
| The method of training I received was reasonable and appropriate for me | | 1 | 1 | 7 |
| The kinds of wheelchair skills taught were reasonable and appropriate for me | | 1 | 3 | 5 |
| The trainer working with me was reasonable and appropriate for me | | | | 9 |
| The expectations for participating in training and practice sessions were manageable and practical | | | 4 | 4 |
| The essential components of the training program were provided as described at the study outset | | | 2 | 7 |
| I was able to perform or improve skills taught in the training program | | | 2 | 7 |
| I did not experience an injury or undue physical or mental stress | | | | 9 |
| The training program was successful in improving my wheelchair skills | | | 1 | 8 |
| Response total (%) | | 2.5 | 16.3 | 81.3 |

for those who met the minimum training time; both values exceeded the MDC target of 3.0%. The mean change score for participants meeting the preferred training dose (5.6 ± 8.4%) was higher than participants meeting the minimum training dose (1.9 ± 1.7%), but did not reach a level of statistical significance ($t = 0.58$, df = 5, $p = 0.59$). The results of the post-treatment questionnaire for EPIC Wheels completers are summarized in Table 5. A total of 98% of question responses were in agreement, with over 81% being "strongly agree".

# DISCUSSION

Overall, the EPIC Wheels study demonstrated a robust protocol that enabled implementation of the intervention in a safe, efficient and acceptable manner. While participant recruitment was challenging, retention and adherence proved to be strong. With respect to process issues, the rate of recruitment and number of participants were below the study targets. Two revisions to the inclusion criteria were made to promote recruitment during the study period: lowering the minimum age from 55 to 50 years and eliminating the maximum period of MWC use. A variety of recruitment strategies were implemented including advertisement in a wide variety of venues (e.g., hospital/rehabilitations centers, health clinics, wheelchair vendors, senior centers, libraries, rehabilitation expos); multiple expositions and engagement with occupational and physical therapists who prescribe wheelchairs; local community newspaper articles and a television interview; telephone invitation via a wheelchair provider program; and hiring an occupational therapist part-time for dedicated recruitment activities. Monthly teleconference meetings were held with study staff at both sites to provide updates, discuss implementation issues, and brainstorm recruitment strategies.

One reason for the low response rate may be related to using a recruitment approach where potential participants were required to initiate contact with the study coordinator, with whom they were unfamiliar. The literature identifies that _passive_ and _opt-in_ recruitment strategies (i.e., where the onus of contact rests with the participant) have

substantially lower success than active and opt-out strategies, where individuals are contacted directly and provide an affirmative response or request no further contact (*Page & Persch, 2013*; *Sygna, Johansen & Ruland, 2015*). Another potential issue may have been MWC users' apprehension over the possibility of allocation to the control group. *Howard et al. (2009)* report that clinicians assisting with recruitment may also have reservations about referring clients who might then be allocated to the control arm of a study. The possibility of allocation to either arm was explicitly conveyed in the consent form; in future, alternative approaches such as avoiding the term "control group" and describing two training program alternatives or using a wait-list approach might be more effective.

The study inclusion criteria also required participants to propel their wheelchair with two hands, precluding a number of user groups such as those who exclusively foot-propel and those with hemiplegia (e.g., post-stroke). The EPIC Wheels training material demonstrated bilateral propulsion strategies, necessitating this inclusion criterion. It was anticipated that, after demonstrating feasibility, additional content could be developed targeting a variety of populations (e.g., hemiplegia, lower limb amputation, male/female specific) as well as devices (e.g., power wheelchairs, scooters). Moving forward, a large-scale RCT should incorporate a variety of content streams within the EPIC Wheels delivery platform to expand the potential recruitment audience.

The number of individuals expressing interest via clinician referral was relatively small, despite considerable effort in marketing the study to practicing therapists. This response is consistent with some other studies in the literature, where authors cite health providers' lack of time in their clinical work, forgetting to bring the study to their clients' attention, concerns about client burden and not prioritizing study recruitment as challenges (*Hubbard et al., 2015*; *Miller et al., 2013*; *Sygna, Johansen & Ruland, 2015*). *Tyson et al. (2015)* used hospital therapists to assist with recruitment and found those who were "gate keepers" (i.e., "pre-screened" clients, rather than providing all eligible clients with the option of participation) had lower recruitment rates; this finding appears to be particularly prevalent when recruiting older adults (*McMurdo et al., 2011*). *Hubbard et al. (2015)* highlight the importance of "buy-in" when including practicing clinicians in recruitment for rehabilitation trials, and propose that clinicians' poor outcome expectations of recruitment, consistent with social cognitive theory, influence their effort and adherence to recruitment strategies. It is also possible that clinicians' outcome expectations for middle-aged and older adults to benefit from wheelchair skills training influenced decisions about passing along study information. We do not know whether any of these factors were relevant for the therapists engaged to assist with recruitment in the EPIC Wheels study. However, given the fact that therapists contacted for this study had identified being involved with middle-aged and older adult MWC users on a regular basis, a better understanding of the factors that impact clinician referral is desirable.

Approximately one-third of eligible individuals declined to participate in the study, exceeding the target parameter of <10%. A variety of reasons were given for not participating, but were primarily related to the distance, cost, or lack of an escort to travel

to the data collection and training sessions. Since all individuals who agreed to participate ultimately provided consent, the rate of refusal was acceptable. Any follow-up study should consider the economic costs of travel for data collection and training session and ensure sufficient resources are available to enable trainees to attend without undue hardship, or provide alternative venues that are accessible and convenient (*Blanton et al., 2006*). The retention rate of 87.5% surpassed the feasibility criteria set. It was anticipated that control group participants would be more inclined to be lost to attrition; however, the only withdrawal was from the EPIC Wheels group and was due to a health condition that prevented continuation in the program. One participant in the control group expressed frustration with his allocation and declined the second training session, but did attend the post-treatment data collection. It appears that, once enrolled, participants were engaged in the study and aside from extenuating circumstances (i.e., health issues) were motivated to complete the study.

The challenges with recruitment experienced in this feasibility study raise concerns about a subsequent clinical trial being sufficiently powered to detect a treatment effect. Suggestions in the preceding paragraphs, such as expanding the program content, could assist in broadening the pool of potential participants. Increasing the number of study sites could augment the total number of participants; however, the speed of recruitment will need to be addressed. The avenues by which the study was advertised were diverse, but relied heavily on the MWC users initiating contact. Middle-aged and older adults often feel overwhelmed during the period of transition to MWC use and are often dealing with a multitude of competing demands (e.g., hospital discharge planning, purchasing adaptive equipment, accessibility issues in the home, other rehabilitation therapies, etc.) (*Giesbrecht, Miller & Woodgate, 2015*). We established positive connections with occupational and physical therapists by providing education sessions, recruitment material, and follow-up inquiries; however, despite general affirmation for the training program, relatively few participants were recruited directly through clinician referral. In order to optimize this recruitment source, it may be helpful to secure an advocate or "champion" within the health care system, and create a formal collaboration with regional health authorities. *Tyson et al. (2015)* advocate for such an approach, suggesting on-site champions can create a culture that identifies recruitment as a role for everyone and maintain enthusiasm over the course of the study. This approach could introduce additional recruitment resources and contextualize the training program as an option that is offered to all eligible MWC users who come in contact with these health systems.

Adherence to treatment was reasonably strong for both study groups, exceeding the target of 85%. Among the EPIC Wheels participants, the rate of adherence to the minimum training time standard (300 min) was above the 85% target, with 57% exceeding the preferred 600 min of training. The patterns of training were quite variable in terms of the number of days per week and duration of practice. Participants typically chose not to practice five days every week, but training sessions duration was consistently appropriate, lasting at least 15 min and not exceeding 75 min. It should be noted that the program software calculates usage based on the start and stop time for each video-based component. Consequently, usage data would likely underestimate the

total time participants spent interacting with the EPIC Wheels program, since additional time would be required to navigate between video components. Participants may have actually had longer "sessions" of training than the data files reflect, and may have been engaged in processing training information or performing wheelchair maneuvers during periods of time between video component activation.

Resource issue demands were acceptable for administration of the intervention, as the trainers were largely able to conduct training sessions in the prescribed time frame, although some flexibility with the length of the second training session should be integrated in a subsequent study. Data collection turned out to be more time-consuming that anticipated, with the majority of both baseline and post-intervention sessions exceeding the expected timeframe. While some participants required more frequent rest breaks between measures, the majority of time was spent administering measures. In particular, three specific tools accounted for over 80 min of time: the WST-C ($37.4 \pm 15.6$ min), the WheelCon ($22.8 \pm 8.0$ min), and the WhOM ($21.1 \pm 16.5$ min). Several strategies might be employed in future to expedite the data collection process and reduce both the tester and participant burden. First, the WheelCon 3.0 is now available in a revised 21-item short-form version, which shows promise as a reliable alternative to the 65-item original test (*Sakakibara, Miller & Rushton, 2015*). Second, future participants might be provided with preparatory questions or information in advance of the data collection sessions. For example, the WhOM is administered as a semi-structured questionnaire and respondents identify up to ten relevant activities they perform using a wheelchair, which can be time-intensive. A third option would be to provide participants with some of the measures to complete in advance and then review them for accuracy and completeness at data collection. Finally, the WST is also available in the WST-Q (questionnaire) version (*Mountain, Kirby & Smith, 2004*), evaluating both capacity (i.e., what can you do) and performance (i.e., what do you do), which is highly correlated with the objective WST-C version used in this study and considerably quicker to administer (~10 min) (*Rushton, Kirby & Miller, 2012*).

Administration time for the HUI was within the expected parameters. There was a statistically significant difference in EPIC Wheels participant HUI scores at baseline and post-treatment, indicating the intervention may have a measurable impact. Given the encouraging results and minimal administration burden, the HUI can be included as an outcome in subsequent trials to measure health-related quality of life and potentially evaluate cost-effectiveness (*Furlong et al., 2001*; *Horsman et al., 2003*).

Regarding management issues, the tablet and related equipment proved to be quite robust with no loss or damage reported and minimal downtime during the training intervention. Transitioning participants into their training program following enrolment was relatively quick, but the average delay was slightly above the targeted value of <10 days. After enrolment, the study statistician was contacted by email to initiate the randomization procedure and respond with the group allocation; this typically required one to three days to process. Booking the initial training session could only be initiated once this was complete and required coordination of schedules between the trainee and trainer, as well as the training space. Given these variables, the processing time

was not unreasonable, but a subsequent study would implement a more expeditious procedure for group allocation and prepare "availability" schedules to offer participants. The trainers found the time provided to be sufficient and the intervention protocol clear, with only a few minor deviations required. These minor adjustments to process were logical and prudent decisions made, which speaks to the need for the study trainers to have sufficient experience and clinical reasoning skills. The modifications made to the treatment protocol were relatively minor, suggesting that the current protocol could be employed in a subsequent trial. Based on the experiences of this feasibility trial, it would be prudent to append a "Frequently Asked Questions" section to the protocol including guidance on potential circumstances where minor deviations might be warranted.

With respect to treatment issues, the feasibility study was conducted without major injury or adverse events occurring during data collection, in-person training, or home training activities. Several situations arose where two participants did experience health issues; these were not incurred due to study-related activities but clearly could impact participation and performance in training and data collection. This is particularly relevant given the target population of middle-aged and older adults whose mobility is compromised due to a health condition, as they are especially vulnerable to concomitant injuries and development of co-morbidities. A large-scale RCT study with the EPIC Wheels program will incorporate strategies to deal with such situations should they arise (e.g., guidelines for suspending or extending training/data collection based upon emergent health events).

While the intent of this study was to establish feasibility, there is evidence to suggest that the recommended treatment dose was sufficient to generate improvement beyond the MDC threshold, and that increasing dosage (i.e., training time) may have additional benefit. The video-based training material in the EPIC Wheels program is approximately 250 min in length, of which 150 min is instructional content. All but one participant met the minimum training requirement of 300 min; however, only four achieved the preferred training goal of 600 min over the study period. Based on these findings, we will recommend participants have flexibility to incorporate tablet-based activities and practice independent of the tablet program, but encourage increasing total practice beyond the minimum standard.

The perception of program benefit for EPIC Wheels participants was strong. All but one participant agreed or strongly agreed with every statement on the post-treatment evaluation form, and the agreement rate of nearly 98% was well above the target feasibility parameter. All EPIC Wheels participants felt that wheelchair skill training was important and eight of nine strongly agreed that the method of training was appropriate and had succeeded in improving their skill level. Only one participant provided a non-agreement response; as a very experienced user, they disagreed that the method of training and types of skills taught were appropriate for them, but still agreed the program had improved their skill level. In summary, EPIC Wheels participants perceived the program to be relevant, appropriate and beneficial.

Based upon the analysis of the feasibility indicators, a full-scale RCT evaluating the clinical impact of the EPIC Wheels intervention is recommended, provided the recommendations for recruitment are implemented. Establishing a collaborative partnership with health care authorities is essential to facilitate on-site champions and implement an active, opt-out recruitment strategy. In addition, expanding and customizing the intervention content would allow access to a larger potential pool of participants. Sufficient financial resources will be required to cover participant transportation costs and Wi-Fi access when necessary. Use of the HUI measure will also allow integration of cost-effectiveness outcomes in a subsequent study.

### Limitations

The indicators used to measure feasibility were established prior to undertaking the study and were based on recommendations from multiple sources within the research literature. However, no standardized format or tool is currently validated for use in medical rehabilitation and consequently the feasibility evaluation was based upon the selection of recommended indicators and assigned target criteria for success. The recruitment challenges and resulting small sample size provided less data upon which to evaluate the feasibility criteria, and limited generalizability of the results, as the study sample may not be representative of the larger population of middle-aged and older adult MWC users. A larger sample might have uncovered additional issues in the data collection and intervention protocols, safety risks, and equipment issues. In particular, individuals who employ a hemiplegic or foot propulsion strategy were excluded; given that 25–75% of individuals who experience a stroke use a wheelchair (*Charbonneau, Kirby & Thompson, 2013*), this is a sizeable population that should be included. The collection of tablet usage and adherence data for the control group participants would have been useful and could potentially have also been compared to the adherence indicator, as was the case with the EPIC Wheels group. Several outcomes had some missing data where the study Tester or Trainer information was not documented. Greater diligence and oversight strategies should be employed in future to ensure a comprehensive and complete data set for all procedures.

## CONCLUSION

This study confirmed the vast majority of feasibility indicators were met or exceeded, with recruitment presenting the greatest challenge. Adherence was generally strong and participants were willing to invest time and effort to achieve the training expectations. Participants' perceived benefit of the intervention was high, even among those with more extensive experience using a MWC. The affirmative findings and recommendations for minor adaptation suggest that a large-scale RCT design is viable, provided a more comprehensive and active recruitment strategy is employed.

## ACKNOWLEDGEMENTS

The authors would like to recognize Andy Kim, Tom Jin and Ian Mitchell for their contributions in developing the EPIC Wheels software and delivery platform.

### Funding
This work was supported by an operating grant from the Canadian Institutes of Health Research (#MOP-123240). The funders had no role in study design, data collection and analysis, decision to publish, or preparation of the manuscript.

### Grant Disclosures
The following grant information was disclosed by the authors:
Canadian Institutes of Health Research: #MOP-123240.

### Competing Interests
The authors declare that they have no competing interests.

### Author Contributions
- Edward M. Giesbrecht conceived and designed the experiments, performed the experiments, analyzed the data, wrote the paper, prepared figures and/or tables.
- William C. Miller contributed reagents/materials/analysis tools, reviewed drafts of the paper, contributed to study and intervention design.

### Human Ethics
The following information was supplied relating to ethical approvals (i.e., approving body and any reference numbers):

This study was approved by the Research Ethics Boards at the University of British Columbia (H12-02043) and the University of Manitoba (H2012:330), as well as the Research Review Committee for regional health authorities at each site.

### Clinical Trial Ethics
The following information was supplied relating to ethical approvals (i.e., approving body and any reference numbers):

This study was approved by the Research Ethics Boards at the University of British Columbia (H12-02043) and the University of Manitoba (H2012:330), as well as the Research Review Committee for regional health authorities at each site.

### Data Availability
The raw data has been provided as Supplemental Dataset Files.

### Clinical Trial Registration
The following information was supplied regarding Clinical Trial registration:

ClinicalTrials.gov #NCT01740635

### Supplemental Information
Supplemental information for this article can be found online at http://dx.doi.org/10.7717/peerj.3879#supplemental-information.

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
