# Peer review of "A randomized control trial feasibility evaluation of an mHealth intervention for wheelchair skill training among middle-aged and older adults"

_PeerJ, doi:10.7717/peerj.3879_

## Round 0.1 · original submission · Minor Revisions

It was great to see that you post for feasibility design on a theoretical framework, in this case one described by Thabane. As suggested by reviewer two, please be careful to ensure that you more clearly describe how the physically data is to be collected. Based on the positive feedback from the two reviewers and myself, I am happy to notify you that you are able to submit a revised version of this manuscript to PeerJ.

Reviewer 1 ·

Basic reporting

no comment

Experimental design

no comment

Validity of the findings

no comment

Additional comments

Authors have done a wonderful work describing their feasibility study which would help them carry out a larger randomized controlled trial in future. The manuscript is very well written in professional English.

I have few minor specific comments which are detailed below:
L40: Please mention if the values in L41 are Mean+/-SD or Mean+/-SEM or something else?
L47: "questionnaire."
L148: Consider replacing "reticence" with something simple.
L219: "was randomly assigned"
L224: "the group"
Figure1: Consider reducing the size of Figure 1
L258: "were 44"
L306: "wheelchair usage might"
Table 2: What is DC1, DC2, HUI. Consider describing them in a footnote of this Table.
L366: "training) were tabulated"
L371: Use the Greek symbol alpha
L390: Why there is n=7 in Table 3's title and not n=9? Also in Table 3, to be consistent consider using 8.0 and 26.0 for the range.
L455: "expositions"
L464: "identifies that"
L482: "Moving forward,"
L516: "frustration with his/her"
L517: "attend the post-treatment"

Reviewer 2 ·

Basic reporting

1. The introduction of this paper reviewed sufficient past related research and have a clear field background for this study.

2. Article structure, figures, tables, and data are provided appropriately.

Experimental design

1. This study which used 2 x 2 factorial design RCT, recruited 18 participants with a retention rate of 94% and practiced their mHelath MWC skills training program for older adults met the good technical standard in experimental design.

2. The method of this study was also described clearly with detail information to replicate.

Validity of the findings

1. The results of EPIC Wheels group post-treatment questionnaire showed that participants had high satisfaction of the program, however, there is a need to state how those questionnaires were distributed and administrated to clarify if participants felt free and comfortable to give their feedback without being afraid of ruining their patient-trainer relationship.

Additional comments

1. According to the definition of WHO and ageing and aged research, most developed world countries have accepted the chronological age of 65 years as a definition of 'elderly' or older person. However, in this study, participants were aged 55 and older. Please reconsider if the title still remind the use of "older adults" or only use the participants who are aged 65 and older.

2. Gender always plays an important role on the level of involving interventions or programs in ageing and aged related studies. Please add gender information and possible demographic background (age distribution, race,...etc.) to understand the overall look of the participants.

---

## Round 0.2 · Minor Revisions

The authors are to be congratulated for attending to almost all of the constructive criticisms from the first round of review. I have the following two areas in which the manuscript can still be improved:

1. The font size in Figure one still appears to large, especially in comparison to Figure two. I therefore suggest you make the font size in Figure one smaller is initially requested by reviewer one.

2. I also do not feel you have quite address the comments from reviewer to regards to the definition and usage term older adult. As you have a minimum age of 50 years to participate in this study, I suggest you used this explicit term (adults aged 50 years or greater) or perhaps middle-aged and older adults throughout the manuscript.

---

## Round 0.3 · accepted · Accept

You have now made all requested amendments to the manuscript and we are happy to advise you that it has now been accepted for publication.